# COPD-Related Mortality before and after Mass COVID-19 Vaccination in Northern Italy

**DOI:** 10.3390/vaccines11081392

**Published:** 2023-08-21

**Authors:** Ugo Fedeli, Veronica Casotto, Claudio Barbiellini Amidei, Andrea Vianello, Gabriella Guarnieri

**Affiliations:** 1Epidemiological Department, Azienda Zero-Veneto Region, 35131 Padua, Italy; ugo.fedeli@azero.veneto.it (U.F.); veronica.casotto@azero.veneto.it (V.C.); claudio.barbielliniamidei@azero.veneto.it (C.B.A.); 2Department of Cardiac-Thoracic-Vascular Sciences and Public Health, Respiratory Pathophysiology Division, University of Padova, 35126 Padua, Italy; andrea.vianello.1@unipd.it

**Keywords:** COPD mortality, epidemiology, COVID-19 vaccines, generalized estimating equation (GEE) models

## Abstract

Background/Objective: Little is known about the impact of the COVID-19 pandemic on mortality from COPD at the population level. The objective was to investigate COPD-related mortality throughout different epidemic waves in Italy before and after the vaccination campaign, which started in late December 2020 and initially targeted the population aged ≥80 years. Methods: Death certificates of residents in Veneto (Northeastern Italy) aged ≥40 years between 2008 and 2021 were analyzed. Age-standardized morality rates were computed for death certificates with any mention of COPD. Generalized estimating equation (GEE) models were fitted to estimate the expected mortality during the pandemic. The results were stratified by age groups of 40–79 and ≥80 years, main comorbidities, and place of death. Results: COPD was mentioned in 3478 death certificates in 2020 (+14% compared to the 2018–2019 average) and in 3133 in 2021 (+3%). Age-standardized mortality rates increased in all age and sex groups in 2020; in 2021, mortality returned to pre-pandemic levels among the elderly but not in the population aged 40–79 years (+6%). GEE models confirmed this differential trend by age. COPD-related mortality peaks were observed, especially in the first pandemic waves, with COVID-19 identified as the underlying cause of death in a relevant proportion (up to 35% in November 2020–January 2021). Mortality with comorbid diabetes and hypertensive diseases slightly increased during the pandemic. Conclusion: COPD-related mortality increased at the beginning of the pandemic, due to deaths from COVID-19. The start of the vaccination campaign was associated with an important decline in COPD-related mortality, especially among the elderly, who first benefited from COVID-19 vaccines. The study findings show the role of mass vaccination in reducing COPD-related deaths during the later phases of the pandemic.

## 1. Introduction

Chronic obstructive pulmonary disease (COPD) represents a well-known risk factor for severe coronavirus disease 2019 (COVID-19). The prevalence of COPD in COVID-19 ranges from 0.70% to 70.6% [1]. A systematic review and meta-analysis including more than 10,000 COVID-19 cases showed that COPD is associated with increased disease severity and progression (RR = 4.20; 95% CI = 2.82–6.25 and RR = 7.48; 95% CI = 1.60–35.05, respectively), intensive care unit admission (RR = 5.61, 95% CI = 2.68–11.76), need for invasive ventilation (RR = 6.53; 95% CI = 2.70–15.84), and a worse overall clinical course (RR = 8.52; 95% CI = 4.36–16.65) [2]. A Chinese retrospective cohort study with 39,420 enrolled subjects with COVID-19 showed that patients with COPD (OR = 1.71; 95% CI = 1.44–2.03) and asthma (OR = 1.45; 95% CI = 1.05–1.98), even after adjusting for age, gender, and other systemic comorbidities, were more likely to require invasive ventilation, be admitted to an intensive care unit, or die within 30 days. Girardin et al. evaluated more than 11,000 patients hospitalized for COVID-19, showing that COPD was the pathology with the most significant increase in mortality (HR = 1.27; 95% CI = 1.02–1.58), while obesity, diabetes, and hypertension were also independent predictors [3]. In a recent systematic review of 10,525 subjects with COPD and COVID-19, the prevalence of death was 40.6% (95% CI, 32.02–49.17) and the odds ratio of mortality among patients with preexisting COPD was calculated at 3.79 [1]. In fact, other respiratory infections, such as influenza and community-acquired pneumonia, are recognized as risk factors for high mortality in COPD patients. In addition, in COPD, factors such as older age and comorbidities, including diabetes and cardiovascular disease are risk factors for high-severity COVID-19 [4].

In spite of such evidence, only sparse data are available on the impact of the pandemic on COPD as a cause of death at the population level, mostly limited to the year 2020. The main problem is that COVID-19 has led to a different classification and distribution of the most frequent and well-known causes of mortality [5]. As a consequence, in order to assess the burden of COPD-related mortality during the pandemic, analyses based on any mention of COPD in death certificates (the so-called multiple causes of death (MCOD) approach) are better suited than statistics limited to the underlying cause of death (UCOD), the single condition selected from all conditions reported on the certificate, following coding rules set by the World Health Organization. According to analyses limited to the UCOD, in 2020, a decrease in mortality from COPD was reported from countries severely affected by the pandemic, such as the US [6], Spain [5], and Scotland and Wales [7]. By contrast, based on MCOD analyses, a 14% increase in the number of COPD-related deaths was registered in 2020 compared to the 2018–2019 average in the Veneto region (Northeastern Italy, 4.9 million inhabitants) [8]; similar analyses carried out in Spain reported a 7.3% increase [9]. According to these latter observations, the long-term declining trend in COPD mortality observed in Europe over recent decades, at least in the male gender [10,11], was abruptly interrupted in 2020.

The current availability of MCOD data spanning beyond the pandemic’s first year allows investigating changes in COPD-related mortality in subsequent phases, characterized by the appearance of new viral variants, the implementation of COVID-19 mass vaccination, and the availability of targeted therapies against COVID-19. In the last two years, numerous vaccines against the SARS-CoV-2 virus have been studied and administered, and many of these have proven effective. In particular, in the context of a randomized phase-3 study, the subgroup analysis relating to the mRNA-1273 vaccine showed significant efficacy both in patients with or without risk factors for severe diseases, such as chronic lung disease [12]. The vaccination campaign in Italy started on 27 December 2020, and initially was administered to healthcare workers, long-term care residents, and people aged over 80 years; thereafter, vaccination was extended to clinically vulnerable subjects and to progressively younger age groups [13]. By the end of March 2021, 74% of the Veneto population aged ≥80 years had received at least one vaccine dose; the corresponding proportion among subjects aged 40–79 years was as low as 13%. By the end of June 2021, a full vaccination coverage was reached among the elderly; in the 40–79 years age bracket, the proportion with at least one vaccine dose was 70%, which increased to 82% by the end of the year.

This study analyzes COPD-related mortality in Veneto throughout the different epidemic waves hitting the region in 2020–2021 by means of the MCOD approach. This study aims to investigate the impact of the vaccination campaign among the population with COPD by assessing changes in COPD-related deaths by age, gender, comorbidities, and place of death, before and after mass vaccination.

## 2. Methods

The Veneto mortality register includes all diseases mentioned in death certificates coded according to the International Classification of Diseases, 10th Revision (ICD-10). Since 2018, the selection of the UCOD has been performed by means of the IRIS software [14]. All deaths of residents in Veneto aged ≥40 years with any mention of COPD (ICD-10 codes J40–J44, J47) were extracted from 1 January 2008 to 31 March 2022. Data for the first trimester of 2022 are provisional, with an estimated coverage of about 97–98%, and were only used to confirm the trends observed at the end of 2021. Selection criteria were the same as adopted in previous studies on COPD-related mortality in Veneto, both in the pre-pandemic period and during the first year of the pandemic [8,11]. The yearly and monthly number of deaths and age-standardized mortality rates with 95% confidence intervals (direct standardization, 2013 European reference population) were computed for any mention of COPD in death certificates (MCOD). Analyses were carried out both in the whole study population and separately by sex and two broad age groups, 40–79 and ≥80 years, chosen based on the different timing of recruitment to the vaccination campaign. The monthly number of COPD-related deaths during the pandemic was also computed based on the selection of COVID-19 as the UCOD (ICD-10 codes U07.1–U07.2), by place of death, and by mention on the death certificate of common comorbidities: hypertensive diseases (ICD-10 codes I10–I13), ischemic heart diseases (I20–I25), cerebrovascular diseases (I60–I69), neoplasms (C00–D48), diabetes (E10–E14), and dementia/Alzheimer’s disease (F01–F03, G30).

Using monthly age standardized COPD-related mortality rates registered from 2008 to 2019, generalized estimating equation (GEE) models were applied to estimate the expected mortality during the pandemic months. Rates were modelled assuming a Gamma distribution with a log-link function and a first-order autoregressive structure. The model included dummy variables for each month (with January as the reference category) to account for the known seasonality of COPD-related deaths; a linear trend across months of the study period was introduced to account for the pre-existing long-term declining trend. Thereafter, observed monthly age-standardized mortality rates were plotted against expected rates, with 95% confidence intervals. The same methodology was applied separately for the populations aged 40–79 and ≥80 years.

## 3. Results

COPD was mentioned in 3478 death certificates in 2020, representing a 14% increase with respect to the 2018–2019 average; this figure decreased to 3133 in 2021, close to numbers registered in the pre-pandemic period. Age-standardized mortality rates in 2020 increased to a greater extent among males and in subjects aged <80 years; in 2021, rates dropped to lower values compared to those observed in pre-pandemic years, except for an excess that persisted among the population <80 years (Table 1).

Figure 1 reports the monthly number of deaths with mention of COPD and the portion of deaths attributed to COVID-19 as the underlying cause. Peaks in COPD-related deaths were registered in correspondence with subsequent COVID-19 epidemic waves. The Veneto region (along with the whole of Northern Italy) was among the first and more severely hit areas involved in the early phases of the pandemic in March–April 2020. In October 2020–January 2021, a second epidemic wave associated with the alpha variant had a larger impact on mortality. After the start of mass vaccination, epidemic waves were registered in March–April 2021 (mostly alpha and gamma variants) and December 2021–January 2022 (initially delta and then omicron variants). COVID-19 was selected as the UCOD in 12% out of overall 6611 COPD-related deaths registered in 2020–2021, with the highest share (35%) observed from November 2020 to January 2021.

Figure 2 plots the monthly age-standardized COPD-related mortality rates registered from January 2020 to March 2022 compared with those forecast by the GEE model, assuming that the declining mortality would have continued through the study period in the absence of the pandemic. A large excess can be observed in March–April 2020 (first epidemic wave) and in November–December 2020 (second wave), while only limited increases in mortality can be detected in the subsequent period. It is worth noting that the observed rates were lower than those predicted both in January–February 2020 and in February 2021. In age-stratified analyses (Figure 3), among the elderly, rates were very close to the expected values after the second wave, while among the population aged 40–79 years, an excess was still observed throughout the year 2021 and in the last epidemic wave of the study period (December 2021–January 2022). 

During the pandemic, a growing share of COPD-related deaths was registered at home (Appendix A); meanwhile, the proportion with mention of diabetes and (limited to 2021) hypertensive diseases slightly increased. A small reduction in COPD-related deaths in nursing homes, and with mention of cerebrovascular diseases/dementia, was observed only in 2021. These findings are confirmed by monthly numbers plotted in Appendix A; peaks in COPD-related deaths reporting also hypertensive diseases, diabetes, and, to a lesser extent, ischemic heart diseases were registered in all epidemic waves, including that of December 2021–January 2022. By contrast, after the first two epidemic waves, no or only minor changes in COPD-related deaths with mention of dementia or cerebrovascular disease were detected. Meanwhile, the sharp increase in deaths in nursing homes registered in the first two epidemic waves was no longer observed after early 2021 (Appendix A).

## 4. Discussion

The increase in COPD-related mortality registered in Northeastern Italy in 2020 was greatly reduced during 2021. Such a reduction was more evident among the elderly, with mortality rates returning towards baseline levels. By contrast, a residual—although smaller—excess mortality persisted in the younger population; in more recent epidemic waves, peaks of mortality were still observed among subjects with both COPD and diabetes/cardiovascular diseases.

During the pandemic, different factors produced contrasting effects on patterns of COPD-related mortality. Among patients testing positive for COVID-19, the presence of COPD was associated with higher mortality, even after adjusting for other comorbidities [15]. Such increased risk of severe COVID-19 has been confirmed also among adult vaccinated COPD patients [16]. In fact, similarly to the influenza virus and the coronaviruses that cause SARS-CoV and MERS-CoV, COVID-19 results in a risk factor for respiratory failure and mortality; among other things, it has been found that about 50% of COPD exacerbations are caused by viral infections; therefore, it can be assumed that SARS-CoV-2 can also cause exacerbations in COPD patients [17]. Furthermore, the incidence of hospitalization and disease severity in patients with pre-existing respiratory diseases, such as COPD, are much higher in patients with COVID-19 than in similar data related to influenza virus [18]. There have been consistent data suggesting that COVID-19 is associated with more severe symptoms and disease course in subjects with COPD in relation to the role played by angiotensin converting enzyme 2 (ACE) [19]. ACE2 is the main host cell receptor for SARS-CoV-2, it has a crucial role in the entry of the virus into the cell and therefore in the spread and induction of the infection. In elderly males and COPD patients, ACE2 levels are upregulated in the epithelium and alveoli of the small airways [20]. This evidence, associated with compromised activity of the innate and adaptive immune responses in COPD patients, can lead to a lower efficacy of respiratory virus clearance [21]; therefore, it is very probable that SARS-CoV-2 can spread and reproduce more easily in the lungs of COPD patients, resulting in more severe COVID-19 [20]. The host genetic predisposition is also a crucial risk factor related to the severity and hypersusceptibility of COVID-19 from numerous genome-wide association studies. HLA-A*11:01, B*51:01 and C*14:02 alleles were significantly associated with worse COVID-19 disease status [22].

On the other hand, lockdown measures with physical distancing and shielding significantly contained the spread and contagiousness of respiratory viruses; studies from different countries found a decline in acute exacerbations of COPD, with reduced hospital admissions and reversal of the usual seasonality pattern [7,23,24,25]. In some countries, this resulted in reduced non-COVID-19 deaths and no excess in overall mortality among the COPD population during the pandemic’s first year [25]. Such an effect was evident in February 2021 in Veneto, with reduced mortality rates due to low influenza circulation compared to what was expected based on seasonality. However, in our study, large peaks in mortality were observed corresponding with the first two COVID-19 epidemic waves, before vaccine availability; this increase was mostly due to the deaths of COPD patients attributed to COVID-19. Such a pattern was similar to that reported for other chronic conditions, with excess mortality estimates for COPD-related mortality being smaller compared to well-known risk factors for severe COVID-19 such as diabetes [26], and larger compared to other conditions such as chronic liver disease [27].

In Italy, it has been estimated at the national level that, from January 2021 to September 2021, the vaccination campaign almost halved COVID-19 deaths in the general population aged ≥80 years. By contrast, the reduction in COVID-19 mortality among younger age groups, recruited later in the campaign and with lower vaccination coverage, was more limited [13]. Such a finding is consistent with global reports focusing on all-cause mortality: in countries that prioritized vaccination among older people, the share of overall deaths among younger subjects increased in the vaccination period compared to the pre-vaccination pandemic period. Such an age shift was not observed in countries with age-independent vaccination policies or with only limited vaccination coverage, providing an indirect but strong support for the effectiveness of the COVID-19 vaccination [28]. According to a population-based study in Veneto, nursing home residents were at much higher risk of COVID-19 mortality and adverse events compared to matched community controls during the first two epidemic waves, before vaccine availability; by contrast during the third wave, after the start of the vaccination campaign, the estimated risk estimates were lower than in controls [29]. Few data are available on changes throughout the different phases of the pandemic in mortality rates associated to specific comorbidity patterns. As regards diabetes-related mortality, an excess was still registered in 2021, but smaller than in 2020; the persistent excess risk during the pandemic’s second year was more pronounced in subjects with both diabetes and hypertensive diseases aged <80 years [26]. All the above reports are consistent with the changes in COPD-related mortality patterns observed throughout the course of the pandemic according to age, comorbidities, and place of death. Peaks in COPD-related deaths among the very elderly, in nursing homes, and in subjects with dementia and cerebrovascular disorders were only observed during the first two epidemic waves. After the beginning of the vaccination campaign, the increase in COPD-related deaths was less pronounced, mostly limited to younger subjects, and to patients affected by hypertensive diseases and other comorbidities specifically associated to an increased risk of COVID-19 adverse outcomes.

Beside mass vaccination, other factors might have contributed to the decline in excess COPD-related mortality observed at a later stage of the pandemic, including the appearance of viral variants with reduced pathogenicity for the lower respiratory tract. However, it must be remarked that the omicron variant became predominant only at the end of the study period, whereas among the elderly, involved in the early phases of the vaccination campaign, no excess in COPD-related deaths was observed after spring 2021. In Italy, a high vaccination coverage of the elderly population was reached in early 2021, in spite of high reported rates of vaccine hesitancy [30]. Within the pandemic context, the importance of increasing health literacy emerged, to obtain a rapid implementation of control measures, including vaccination, requiring the collective compliance of all individuals [31]. It is also relevant to underline the positive role played by the concomitant campaign for anti-flu vaccination in the autumn. From the results of studies that evaluated the correlation between influenza vaccination, susceptibility to SARS-CoV-2 infection, and COVID-19 outcomes, influenza vaccination was protective and associated with a lower risk of 60-day mortality in COVID-19 patients (OR = 0.2; 95% CI = 0.08–0.51) [32]. In a study conducted in the United Kingdom, influenza vaccination was associated with a lower probability of death from all causes in COVID-19 patients (OR = 0.76; 95% CI = 0.64–0.90) [16]. The molecular mechanisms hypothesized by the authors were the action of the influenza virus at the level of pulmonary alveolar cells, leading to an increase in the expression of the ACE2 receptor, which could worsen a subsequent SARS-CoV-2 infection. Therefore, prevention of influenza through vaccination can reduce the viral load and severity of COVID-19. Finally, another factor that probably contributed to limiting the excess mortality from COPD is the clinical efficacy of corticosteroids against severe forms of COVID-19, due to their anti-inflammatory and immunomodulatory properties. Epidemiological data have not demonstrated an increased risk of worse COVID-19 outcomes with ICS use beyond that due to the severity of the comorbidity [33].

The main study limitation is represented by the accuracy of death certificates, which is known to be affected by a large underreporting of COPD and by the absence of additional information on the severity of the disease and on smoking habits. Cigarette smoking is known to be a major risk factor for the development of COPD and has recently been identified as a predisposing factor for SARS-CoV-2 infection [34]. In this case, the role played by the ACE2 receptor is also highlighted; in fact, recent studies have shown that exposure to smoke upregulates lung levels of ACE2, facilitating entry to SARS-CoV-2 and therefore progression of infection [35]. Several studies have suggested that chronic low levels of oxidative stress and inflammation, combined with changes caused by a viral infection, may be responsible for the most severe forms of COVID-19 [36].

Routine mortality statistics largely underestimate the burden of COPD-related deaths for different reasons: underdiagnosis of COPD in the general population; underreporting of known disease in death certificates; selection of a different UCOD in certificates with mention of COPD. As a consequence, sensitivity of death certificates has been found to be very low (about 40%, even in the severe stages) especially in population-based studies where the disease was assessed by screening with lung function testing [37,38], and higher in studies carried out among treated COPD patients [39]. Underreporting, however, is unlikely to have changed significantly across the study period; therefore, estimates of variations through subsequent epidemic waves were unlikely to be affected by this issue. Moreover, the MCOD approach at least addresses one of the mechanisms of underestimation, e.g., selection of a different UCOD, and it is more suitable for assessing time trends [8]. Only MCOD-based analyses can uncover the full burden of COPD-related deaths during the pandemic: in Veneto, analyses limited to the UCOD did not reveal any excess in mortality from COPD during 2020 [8]. Reports from Spain confirmed such a discrepancy: compared to the 2018–2019 average, in 2020, the number of deaths attributed to COPD as the UCOD declined by 10.5%, whereas using the MCOD approach, a 7.3% increase could be observed [9]. It must be remarked that when both COPD and COVID-19 are mentioned in death certificates, the latter is more likely to be selected as the UCOD, due to rules set by the WHO [40]. In view of the above, MCOD analyses must be regarded as the standard methodological approach for evaluating COPD as a cause of death during the pandemic. Moreover, MCOD can provide evidence about the role of comorbidities in increasing the risk of death during epidemic waves. The present study investigated only the more commonly reported comorbidities; however, during the pandemic, among COPD-related deaths, an increase in the mention of other conditions such as obesity was registered, which are less likely to be reported in death certificates but strongly associated with severe COVID-19.

In view of the above, the main strength of this study is the availability of MCOD data spanning multiple epidemic waves, before and after the implementation of the vaccination campaign. With this approach, the present report is the first to describe the increase in COPD-related mortality during the pandemic before the vaccination campaign and the return towards baseline levels after mass vaccination.

## 5. Conclusions

This study has shown how COPD was associated with increased mortality in the first phase of the pandemic, largely accounted for by deaths attributed to COVID-19. After the beginning of the COVID-19 vaccination campaign, an important reduction in COPD-related mortality was observed compared to that registered in the first year of the pandemic. Mortality rates returned to pre-pandemic levels among people ≥80 years, the first who benefited from COVID-19 vaccines. The study findings show the role of mass vaccination in reducing COPD-related deaths during the later phases of the pandemic.

## Figures and Tables

**Figure 1 vaccines-11-01392-f001:**
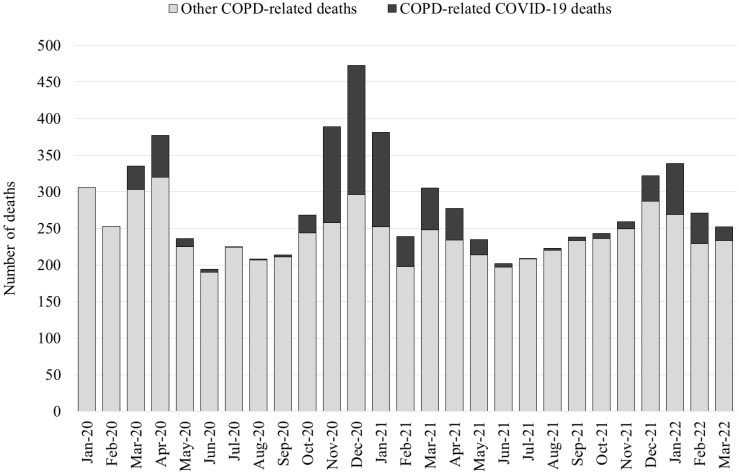
Monthly number of COPD-related deaths, with and without COVID-19 as the underlying cause.

**Figure 2 vaccines-11-01392-f002:**
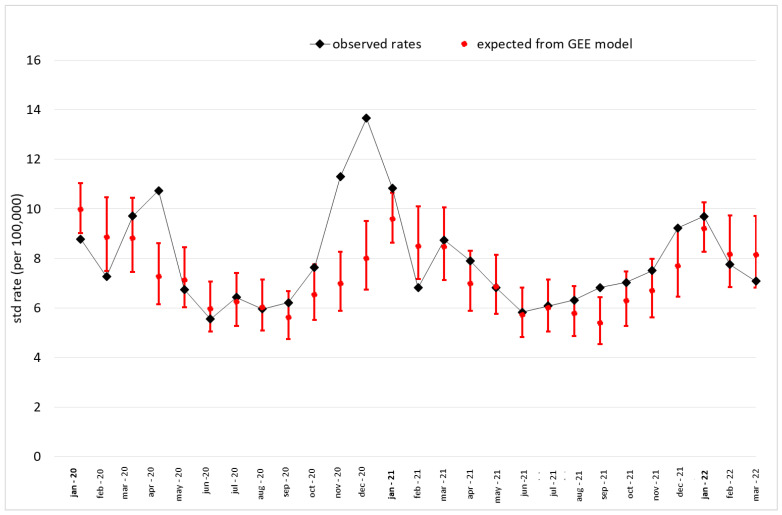
Observed monthly age-standardized COPD-related mortality rates throughout the pandemic period, compared with expected values (and 95% confidence intervals) obtained by a generalized estimating equation model applied to 2008–2019 data.

**Figure 3 vaccines-11-01392-f003:**
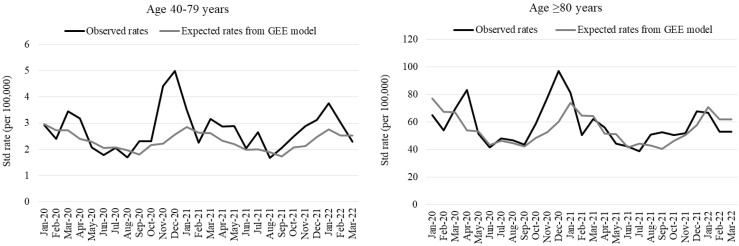
Observed monthly age-standardized rates of COPD-related mortality compared to expected figures obtained using a generalized estimating equation model applied to 2008–2019 data: population aged 40–79 and ≥80 years.

**Table 1 vaccines-11-01392-t001:** COPD-related deaths by sex and age groups (40–79 years; ≥80 years), with the percentage change in 2020 and 2021 compared to the 2018–2019 average.

	Average 2018–2019	2020	2021	Percentage Change	
				2020 vs. 2018–2019 Average	2021 vs. 2018–2019 Average
*All COPD-related deaths*					
Number of deaths	3055	3478	3133	14%	3%
Age-std rates × 10^5^ * (95% CI)	91.1 (88.8–93.4)	100.0 (96.7–103.4)	89.9 (86.8–93.2)	10%	−1%
*Age group*					
40–79 years, *n* deaths	783	887	826	13%	5%
40–79 years, std rates × 10^5^ (95% CI)	29.9 (28.4–31.4)	33.6 (31.4–35.9)	31.6 (29.5–33.8)	12%	6%
≥80 years, *n* deaths	2272	2591	2307	14%	2%
≥80 years, std rates × 10^5^ (95% CI)	678.8 (659.2–698.9)	737.1 (708.9–766.1)	649.8 (623.5–676.9)	9%	−4%
*Sex*					
Males, *n* deaths	1719	2024	1812	18%	5%
Males, std rates × 10^5^ (95% CI)	145.8 (140.8–150.9)	159.8 (152.7–167.1)	142.1 (135.4–148.9)	10%	−3%
Females, *n* deaths	1336	1454	1321	9%	−1%
Females, std rates × 10^5^ (95% CI)	61.1 (58.8–63.5)	64.8 (61.4–68.3)	59.6 (56.3–63.0)	6%	−3%

* Truncated age-standardized rates (≥40 years) using the European population in 2013 as a standard.

## Data Availability

The data are available for reproduction of results on request from the corresponding author.

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
