# Peer review of "COPD-Related Mortality before and after Mass COVID-19 Vaccination in Northern Italy"

_vaccines, 2023, doi:10.3390/vaccines11081392_

Round 1

Reviewer 1 Report

Great analysis of excess COPD death overlapping with SARS-CoV-2 and vaccination. Kindly note estimated percent of vaccinated population in Italy by Summer of 2021 and winter 2021-22. Excess COPd death were near 10%.

This percentage of excess death due to diabetes alone as cited in references should also be state with absolute number in winter of 2020-21 and percent increase for context. This is different than COPD with comorbid diabetes as noted in supplement

Author Response

Dear Reviwer1,

plese see our answers to your comments below:

.Comment 1. Great analysis of excess COPD death overlapping with SARS-CoV-2 and vaccination. Kindly note estimated percent of vaccinated population in Italy by summer of 2021 and winter 2021-22. Excess COPD death were near 10%.

Answer. We thank the referee for the positive consideration given to our work and for possibility to clarify this important point. Vaccination rates in the Veneto region at different time points through 2021 and by age group have been added to the revised manuscript

Comment 2. This percentage of excess death due to diabetes alone as cited in references should also be state with absolute number in winter of 2020-21 and percent increase for context. This is different than COPD with comorbid diabetes as noted in supplement

Answer. The increase in mortality from diabetes, a known risk factor for severe COVID-19 diseases and mortality, has been added to the Discussion to provide more context to study results

Reviewer 2 Report

Title: Impact of the COVID-19 Pandemic and Vaccination Campaign on COPD-Related Mortality in Italy: A Population-Level Analysis

Abstract:

The abstract provides a concise summary of the study, highlighting the background, objective, methods, and key results. It effectively communicates the main findings of the research. However, some improvements can be made to enhance clarity and readability. The abstract should clearly state the significance of the study and its implications for public health. Additionally, it would be helpful to include the timeframe of the vaccination campaign and the specific population groups targeted.

Introduction:

The introduction provides a comprehensive background on the association between COPD and severe COVID-19, emphasizing the risk factors and comorbidities associated with COPD. It also mentions the limited available data on COPD-related mortality during the pandemic and highlights the importance of analyzing COPD mortality based on death certificates with any mention of COPD (MCOD approach) rather than focusing solely on the underlying cause of death (UCOD). However, the introduction would benefit from a clear statement of the research objectives and research questions that the study aims to address.

Methods:

The methods section provides a clear description of the study design, data collection, and statistical analysis. The use of the Veneto mortality register and the inclusion criteria for COPD-related deaths are appropriate. The application of generalized estimating equation (GEE) models to estimate mortality rates during the pandemic is a suitable approach. However, more specific details on the GEE models, including the covariates and model assumptions, would strengthen the methodology section.

Results:

The results section presents the key findings of the study in a logical and organized manner. It effectively communicates the increase in COPD-related mortality during the pandemic and the subsequent reduction following the initiation of the vaccination campaign. The inclusion of mortality rates stratified by age, gender, comorbidities, and place of death provides valuable insights. However, it would be beneficial to include statistical measures of uncertainty, such as confidence intervals, for the reported mortality rates.

Discussion:

The discussion interprets the study findings in the context of previous research and provides insights into the factors influencing COPD-related mortality during the pandemic. The discussion effectively addresses the contrasting effects of lockdown measures and vaccination on mortality patterns. It also highlights the importance of comorbidities and age in the observed mortality trends. However, the discussion could be strengthened by discussing the potential mechanisms underlying the observed reductions in COPD-related mortality following the vaccination campaign.

Conclusion:

The conclusion provides a succinct summary of the study findings and their implications. It effectively communicates the reduction in COPD-related mortality following the initiation of the COVID-19 vaccination campaign, particularly among individuals aged 80 years and older. However, the conclusion could be improved by emphasizing the significance of these findings in the context of public health and the potential benefits of widespread vaccination in reducing the burden of COPD-related mortality.

Overall, this manuscript provides valuable insights into the impact of the COVID-19 pandemic and vaccination campaign on COPD-related mortality in Italy. The study design, data analysis, and presentation of results are generally sound. However, some sections could benefit from minor revisions to enhance clarity, provide additional details, and strengthen the discussion of the findings.

good english

Author Response

Dear Reviwer2,

please see our answers to your comments below:

Comment 1. Abstract: The abstract should clearly state the significance of the study and its implications for public health. Additionally, it would be helpful to include the timeframe of the vaccination campaign and the specific population groups targeted.

Answer. The timeframe of the vaccination campaign has been added to the Abstract to underline the main implication of the study for public health, namely the impact of vaccination in reducing COPD-related mortality during the second phase of the pandemic.

Comment 2. Introduction: The introduction would benefit from a clear statement of the research objectives and research questions that the study aims to address.

Answer. At the end of the introduction, research questions and objective have been stated more clearly .

Comment 3. Methods: More specific details on the GEE models, including the covariates and model assumptions, would strengthen the methodology section.

Answer. GEE models applied to COPD-related mortality rates are now more clearly explained in the Methods section of the revised manuscript

Comment 4. Results: It would be beneficial to include statistical measures of uncertainty, such as confidence intervals, for the reported mortality rates.

Answer. Confidence intervals for mortality rates have been added to the revised Table 1

Comment 5. Discussion: The discussion could be strengthened by discussing the potential mechanisms underlying the observed reductions in COPD-related mortality following the vaccination campaign.

Answer. The Discussion section has been expanded as regards possible mechanisms of reduced COPD-related mortality, both associated to the vaccination campaign and other factors (e.g., change in viral variants)

Comment 6. Conclusion: The conclusion the conclusion could be improved by emphasizing the significance of these findings in the context of public health and the potential benefits of widespread vaccination in reducing the burden of COPD-related mortality.

Answer. The Conclusion has been rephrased to emphasize the impact of widespread vaccination in reducing the burden of COPD-related mortality

Reviewer 3 Report

Dear colleague,

Thank you for the kind invitation to review the manuscript. The manuscript cover important trends related to copd mortality during the covid-19 pandemic. 

Introduction

- Some elaboration should be made regards to data from other international populations before tying to the population on hand

- Some discussion should be made regarding epidemiology of copd in italy and any epidemiological differences with other population to warrant its special focus for this study 

Methods 

- what were the exclusion criteria for the study?

- Do the authors have more recent data i.e for whole of 2022 

-> Will be helpful to do a full analyses 

Discussion

- what are the main implications of the study? 

- Are there any research gaps, unique findings from the study that differ from international studies?

- What are the authors' thoughts regarding health literacy and its interaction with copd related mortality during the covid-19 pandemic. 

-> Studies have shown poor health literacy rates are high especially during the pandemic

-> Cite: https://pubmed.ncbi.nlm.nih.gov/37459004/

Minor comments

- Will be important to discuss the importance of vaccine hesitancy among COPD population in the discussion in view of their association with increased mortality in covid-19 pandemics

-> This is especially so as there are high rates of vaccine hesitancy globally

-> please cite: https://pubmed.ncbi.nlm.nih.gov/34452026/

as above

Author Response

Dear Reviewer3,
please see our answers to your comments below:

Comment 1. Introduction - Some elaboration should be made regards to data from other international populations before tying to the population on hand

Answer. Data on mortality from COPD during the pandemic from other countries (based on the UCOD) are now briefly summarized in Introduction of the revised manuscript.

Comment 2. Introduction -Some discussion should be made regarding epidemiology of COPD in Italy and any epidemiological differences with other population to warrant its special focus for this study

Answer. Pre- pandemic mortality trends of COPD in Europe and Italy are now cited in Introduction.

Comment 3. Methods - what were the exclusion criteria for the study?

Answer. No exclusion criteria was applied to mortality records, except for residency outside the study area and age < 40 years.

Comment 4. Methods - Do the authors have more recent data i.e for whole of 2022 - Will be helpful to do a full analyses

Answer. Data for the year 2022 are provisional, and were only used to provide a picture of the omicron wave in December 2021-January 2022. Nonetheless, this is the first study to investigate COPD-related mortality through the first two years of the pandemic, including the implementation of COVID-19 mass vaccination.

Comment 5. Discussion - what are the main implications of the study? - Are there any research gaps, unique findings from the study that differ from international studies?

Answer. The unique findings of the study are now explicitly stated in the Discussion section, at the end of the paragraph on study limitations and strengths.

Comment 6. Discussion - - What are the authors' thoughts regarding health literacy and its interaction with copd related mortality during the covid-19 pandemic.  - Studies have shown poor health literacy rates are high especially during the pandemic - Cite: https://pubmed.ncbi.nlm.nih.gov/37459004/

Answer. The importance of increasing health literacy in order to reach a rapid implementation of pandemic control measures – included vaccination- is cited in Discussion of the revised paper, and the suggested reference has been added.

Comment 7. Minor comments - Will be important to discuss the importance of vaccine hesitancy among COPD population in the discussion in view of their association with increased mortality in covid-19 pandemics - This is especially so as there are high rates of vaccine hesitancy globally - please cite: https://pubmed.ncbi.nlm.nih.gov/34452026/https://pubmed.ncbi.nlm.nih.gov/34452026/

Answer. The importance of vaccine hesitancy has been added to the Discussion of the revised manuscript, along with the suggested reference. In spite of high rates of vaccine hesitancy reported from high income countries, including Italy, the high vaccination rates rapidly reached among the elderly population contributed to the study finding of a drop in excess COPD-related deaths during the pandemic’s second year.

Reviewer 4 Report

Reviewer comments on manuscript

 COPD-RELATED MORTALITY BEFORE AND AFTER MASS 2 COVID-19 VACCINATION IN NORTHERN ITALY 3

Ugo Fedeli 1, Veronica Casotto 1, Claudio Barbiellini Amidei 2, Andrea Vianello 3 and Gabriella Guarnieri 3*

General comments:

The manuscript is very well written and has a rigorous approach to achieving the specific aims as stated in the narrative. This is one of a series of articles by the primary author Dr Fedeli who is a respected Italian Epidemiologist who has contributed in a significant way in many areas of clinical epidemiology.

The statistical methodology is well established but complex and reported in other publications.

One of the focuses of his work based on reading his other articles is the issue of the recording of the cause of death in the format of listing of a primary cause of death without incorporating other important co morbidities. 

This is very significant as our global population ages and accumulates multiple co morbidities we need to recognize the attributable risks of these various conditions to the ultimate cause of death.

As an academic specialist in Endocrinology in the US for over 32 years the reviewer has completed hundred of death certificate and I can only agree with the concerns raised. Careful consideration on how other co-morbidities contribute in a meaningful way to the “immediate cause of death” are very idiosyncratic much like the accuracy of ICD-10 codes into the medical record.

The immediate cause of death is never Covid, COPD or diabetes. It’s usually respiratory failure, uni or multi organ failure, sepsis with organ failure, cardiovascular event- MI arrythmia heart failure, acute renal failure, or electrolyte imbalance. 

However focusing on one diagnosis such as COPD or COVID as the immediate cause of death rather than as the major contributor to the pathophysiological process to leading to the ultimate immediate cause continues to be a challenge globally

The current manuscript appears to be an extension of his earlier published work.

I do have some questions and I have cut and pasted statements from the manuscript:

In spite of such evidence, only sparse data are available on the impact of the pandemic on COPD-related mortality, mostly limited to the year 2020”

This statement I believe is not accurate based on my review of the existing published data accessed July 2023. Granted severe COVID infections have literally disappeared from our US Medical centers thankfully to date.

1.      Uruma Y, Manabe T, Fujikura Y, Iikura M, Hojo M, Kudo K. Effect of asthma, COPD, and ACO on COVID-19: A systematic review and meta-analysis. PLoS One. 2022 Nov 1;17(11):e0276774. doi: 10.1371/journal.pone.0276774. PMID: 36318528; PMCID: PMC9624422.

2.      Shi L, Wang Y, Han X, Wang Y, Xu J, Yang H. Comorbid asthma decreased the risk for COVID-19 mortality in the United Kingdom: Evidence based on a meta-analysis. Int Immunopharmacol. 2023 Jul;120:110365. doi: 10.1016/j.intimp.2023.110365. Epub 2023 May 22. PMID: 37224652; PMCID: PMC10201319.

3.      Hippisley-Cox J, Coupland CA, Mehta N, Keogh RH, Diaz-Ordaz K, Khunti K, Lyons RA, Kee F, Sheikh A, Rahman S, Valabhji J, Harrison EM, Sellen P, Haq N, Semple MG, Johnson PWM, Hayward A, Nguyen-Van-Tam JS. Risk prediction of covid-19 related death and hospital admission in adults after covid-19 vaccination: national prospective cohort study. BMJ. 2021 Sep 17;374:n2244. doi: 10.1136/bmj.n2244. Erratum in: BMJ. 2021 Sep 20;374:n2300. PMID: 34535466; PMCID: PMC8446717.

4.      Meza D, Khuder B, Bailey JI, Rosenberg SR, Kalhan R, Reyfman PA. Mortality from COVID-19 in Patients with COPD: A US Study in the N3C Data Enclave. Int J Chron Obstruct Pulmon Dis. 2021;16:2323-2326. Published 2021 Aug 13. doi:10.2147/COPD.S318000

“The main study limitation is represented by the accuracy of death certificates, known 191 to be affected by a large underreporting of COPD”

Despite this statement which the reviewer totally agrees-There has been no rigorous attempt to identify this critical deficit. What is the percentage of “under reporting” and as such how does this affect the outcome and conclusions?

“Conclusion

This study has shown how COPD was associated with increased mortality in the first 209 phase of the pandemic, largely accounted by deaths attributed to COVID-19. After the beginning of the COVID-19 vaccination campaign, an important reduction in COPD-re lated mortality was observed compared to that registered in the first year of the pandemic. 212 Mortality rates returned to pre-pandemic levels among p_e_o_p_l_e_ _≥8_0_ _y_e_a_r_s_,_ _t_h_e_ _f_i_r_s_t_ _w_h_o_ _213 benefited from COVID-19 vaccines.”

1.     The reviewer agrees that the results support this conclusion but there is NO non-COPD control group to compare mortality. Is the increased mortality just consistent with the same increased mortality in a general non-COPD population in this region of Italy?

2.     Although I agree with the important effects of vaccination at lowering severe disease the “phases” of the pandemic were largely driven by the viral mutations of the SARS CoV2 virus. The initial phase was alpha, second delta and then omicron. There are numerous publications that propose that omicron is more infectious BUT does not cause the severe lower respiratory damage of delta. One paper suggest omicron is largely an upper respiratory pathogen compared to delta which infects the lower bronchial area. Therefore, is the decrease in the mortality in the later stage just a reflection of omicron?

3.     Vaccination has clearly impacted the incidence of severe disease globally. Do the authors have any vaccination rate data in the cohort?

4.     COPD is a broad term including asthma, emphysema, environmental exposure (asbestosis) genetic conditions such as cystic fibrosis. Do the authors have access to more granular information on these causes of COPD or does it make no difference.

5.      Cigarette smoking with COPD significantly increases covid complications – any information related to this risk?

6.     Obesity at least in the US is a very important covariant but I don’t see that in the analysis?

In summary the authors have largely replicated and validated outcomes in other global locations and the results are highly predictable and they are regional specific.

The reviewer is not convinced there is any thing either novel or more clinically practical beyond already published studies. I would challenge the authors to refute my conclusions and provide more compelling arguments they will add to the existing body of literature.

Author Response

Dear Reviewer4,
please see our answers to your comments below:

Comment 1. “ In spite of such evidence, only sparse data are available on the impact of the pandemic on COPD-related mortality, mostly limited to the year 2020” This statement I believe is not accurate based on my review of the existing published data accessed July 2023. Granted severe COVID infections have literally disappeared from our US Medical centers thankfully to date.  1.      Uruma Y, Manabe T, Fujikura Y, Iikura M, Hojo M, Kudo K. Effect of asthma, COPD, and ACO on COVID-19: A systematic review and meta-analysis. PLoS One. 2022 Nov 1;17(11):e0276774. doi: 10.1371/journal.pone.0276774. PMID: 36318528; PMCID: PMC9624422. 2.      Shi L, Wang Y, Han X, Wang Y, Xu J, Yang H. Comorbid asthma decreased the risk for COVID-19 mortality in the United Kingdom: Evidence based on a meta-analysis. Int Immunopharmacol. 2023 Jul;120:110365. doi: 10.1016/j.intimp.2023.110365. Epub 2023 May 22. PMID: 37224652; PMCID: PMC10201319. 3.      Hippisley-Cox J, Coupland CA, Mehta N, Keogh RH, Diaz-Ordaz K, Khunti K, Lyons RA, Kee F, Sheikh A, Rahman S, Valabhji J, Harrison EM, Sellen P, Haq N, Semple MG, Johnson PWM, Hayward A, Nguyen-Van-Tam JS. Risk prediction of covid-19 related death and hospital admission in adults after covid-19 vaccination: national prospective cohort study. BMJ. 2021 Sep 17;374:n2244. doi: 10.1136/bmj.n2244. Erratum in: BMJ. 2021 Sep 20;374:n2300. PMID: 34535466; PMCID: PMC8446717. 4.      Meza D, Khuder B, Bailey JI, Rosenberg SR, Kalhan R, Reyfman PA. Mortality from COVID-19 in Patients with COPD: A US Study in the N3C Data Enclave. Int J Chron Obstruct Pulmon Dis. 2021;16:2323-2326. Published 2021 Aug 13. doi:10.2147/COPD.S318000

Answer. The sentence has been rephrased: data on COPD as a cause of death are almost completely lacking at the population level; the few available figures might be misleading due to the competing role of COVID-19 as the underlying cause. By contrast, many papers are available on the role of COPD as a comorbidity in increasing the risk of severe COVID-19; the Introduction and the Discussion section have been greatly expanded to better address this issue, and the suggested (and other) references have been added to the revised manuscript.

Comment 2. “The main study limitation is represented by the accuracy of death certificates, known to be affected by a large underreporting of COPD” Despite this statement which the reviewer totally agrees-There has been no rigorous attempt to identify this critical deficit. What is the percentage of “under reporting” and as such how does this affect the outcome and conclusions?

Answer. Available data on the accuracy of reporting COPD in death certificates have been added to the Discussion of the revised manuscript. Underreporting, however, is unlikely to have changed significantly across the study period; therefore, we believe that estimates on variations through subsequent epidemic waves of COPD-related mortality were not affected by this issue. Furthermore, compared to other studies focusing on COPD deaths based only on the underlying cause of death, the use of a MCOD approach mitigated underestimation of mortality from COPD, that during the pandemic was affected as well as by underreporting also by COVID-19 acting as a strong competing underlying cause of death.  

Comment 3. “Conclusion This study has shown how COPD was associated with increased mortality in the first phase of the pandemic, largely accounted by deaths attributed to COVID-19. After the beginning of the COVID-19 vaccination campaign, an important reduction in COPD-related mortality was observed compared to that registered in the first year of the pandemic. Mortality rates returned to pre-pandemic levels among people ≥80 years, the first who benefited from COVID-19 vaccines.” The reviewer agrees that the results support this conclusion but there is NO non-COPD control group to compare mortality. Is the increased mortality just consistent with the same increased mortality in a general non-COPD population in this region of Italy?

Answer. More data on increased mortality related to other comorbidities during the first phase of the pandemic have been added to the Discussion section. For some conditions (e.g., diabetes), the increase in mortality was larger; for others (e.g., chronic liver diseases) was smaller. However, it must be remarked that the study is a time sere analysis of COPD-related mortality rates: the pre-pandemic period represents the control period used to assess the impact of the pandemic.

 Comment 4. Although I agree with the important effects of vaccination at lowering severe disease the “phases” of the pandemic were largely driven by the viral mutations of the SARS CoV2 virus. The initial phase was alpha, second delta and then omicron. There are numerous publications that propose that omicron is more infectious BUT does not cause the severe lower respiratory damage of delta. One paper suggest omicron is largely an upper respiratory pathogen compared to delta which infects the lower bronchial area. Therefore, is the decrease in the mortality in the later stage just a reflection of omicron?

Answer. We agree that other factors, including viral variants, influenced changes in COPD-related mortality during the pandemic. This has been added to the Discussion of the revised manuscript. However, it must be remarked that the omicron variant became predominant in Italy only in December 2022; the drop in COPD-related mortality rates among subjects aged ≥80 years was already evident starting from February-March 2022.

Comment 5. Vaccination has clearly impacted the incidence of severe disease globally. Do the authors have any vaccination rate data in the cohort?

Answer. Vaccination rates by age in the Veneto region have been added to the revised manuscript, to allow a better interpretation of study findings.

Comment 6. COPD is a broad term including asthma, emphysema, environmental exposure (asbestosis) genetic conditions such as cystic fibrosis. Do the authors have access to more granular information on these causes of COPD or does it make no difference.

Answer. Deaths from asthma (ICD-10 J45-J46) were not included in the selection of mortality records. Most of COPD-related deaths were coded as unspecified COPD (J44.x), so analyses by specific subtypes (e.g. predominant emphysema) were not feasible.

Comment 7. Cigarette smoking with COPD significantly increases covid complications – any information related to this risk?

Answer. Data on cigarette smoking are not available form mortality records; the impact of smoking on COVID complications is now discussed in the revised manuscript.

Comment 8.  Obesity at least in the US is a very important covariant but I don’t see that in the analysis?

Answer. We agree that obesity is an important comorbidity that is specifically associated to an increased risk of severe COPD. However, it must be remarked that the issue of underreporting in death certificates, already relevant for COPD, is even more severe for obesity, and this remains a study limit explicitly stated in the revised manuscript. As an example, obesity was mentioned in only 2.5% of COPD-related deaths in pre-pandemic years; this proportion sharply increased to 3.9% in 2020 and 3.4% in 2021. We do not report data on obesity in Supplementary Table 1 and Supplementary Figure 1, where we chose to show only the most commonly reported comorbidities.

Comment 9.  In summary the authors have largely replicated and validated outcomes in other global locations and the results are highly predictable and they are regional specific.  The reviewer is not convinced there is any thing either novel or more clinically practical beyond already published studies. I would challenge the authors to refute my conclusions and provide more compelling arguments they will add to the existing body of literature.

Answer. We agree that study results might not be considered unexpected in theory (increase in COPD-related mortality during the pandemic before the vaccination campaign, return towards baseline levels after mass vaccination). However, although quite surprisingly, this is the first study to assess this effect at the population level. This compelling argument is now explicitly stated at the end of the Discussion of the revised manuscript.

Round 2

Reviewer 3 Report

nil comments

require english editing

Reviewer 4 Report

The reviewer thanks the authors for the thoughtful and critical consideration of my comments and I agree with the excellent edits